# Changes in Physical Performance among Community-Dwelling Older Adults in Six Years

**DOI:** 10.3390/ijerph20085579

**Published:** 2023-04-19

**Authors:** Darlise Rodrigues dos Passos Gomes, Leonardo Pozza Santos, Maria Cristina Gonzalez, Edgar Ramos Vieira, Renata Moraes Bielemann

**Affiliations:** 1Post-Graduate Program in Food and Nutrition, Federal University of Pelotas, Pelotas 96075-630, Brazil; cristina.gonzalez@pbrc.edu (M.C.G.); rbielemann.fn@ufpel.edu.br (R.M.B.); 2Department of Nutrition, Federal University of Pelotas, Pelotas 96075-630, Brazil; lpozza.fn@ufpel.edu.br; 3Department of Physical Therapy, Florida International University, Miami, FL 33199, USA; evieira@fiu.edu; 4Post-Graduate Program in Epidemiology, Federal University of Pelotas, Pelotas 96020-220, Brazil

**Keywords:** aging, walk tests, community-dwelling

## Abstract

Changes in physical performance (PP) occur with aging, and understanding the magnitude of these changes over time is important. This study evaluated changes in Gait Speed (GS) and Timed Up and Go (TUG) performance and their association with related factors among community-dwelling older adults over a period of five to six years. A cohort study of 476 older adults with baseline assessment conducted in 2014 and reassessment in 2019–2020 was carried out. Associations between PP changes over time and sociodemographic, behavioral, and health variables were evaluated using mixed linear models. Approximately 68% of the participants declined PP; 20% had no relevant change in GS and 9% in TUG time (sustained PP); 12% increased GS, and 23% decreased TUG time (improved PP). Being male (*p* = 0.023), living without a partner/separated (*p* = 0.035), higher education (*p* = 0.019), and alcohol consumption in the prior month (*p* = 0.045) were associated with decreased GS, while older age (*p* < 0.001), having lower socioeconomic status (*p* < 0.004), physical inactivity (*p* = 0.017), and being overweight (*p* = 0.007) were associated with increased TUG time. PP declined for most participants. Factors most strongly associated with PP decline are non-modifiable. The high prevalence of PP decline over time signals the importance of including physical tests in yearly health assessments.

## 1. Introduction

Physical performance (PP) is a critical health indicator for older adults [1]. Decline in PP is associated with frailty, sarcopenia, disability, cognitive decline, falls, hospitalization, and higher mortality risk [1,2,3,4]. PP changes occur with aging as a dynamic process. The start and magnitude of decline differ between individuals, type of PP test, and setting [5,6,7]. Aging-related declines in cardiovascular, musculoskeletal, and neuromuscular systems’ function result in progressive loss of muscle mass and strength. Sociodemographic, lifestyle (e.g., diet quality and physical activity levels), depressive symptoms, multimorbidity, and nutritional status contribute to these changes [2,3,6,8,9,10]. However, further understanding of the factors associated with PP changes over time in community-dwelling older adults is needed to plan optimal prevention strategies.

Recent studies assessing PP trajectories are based on the categorization of physical performance tests (PPTs) according to cut-off points, which can be useful to identify mobility disorders. However, even if individuals are above the low PP threshold, declines in performance have clinical and practical implications, and identifying these changes can contribute to early diagnosis and implementation of preventive measures in a timely manner [7,11,12,13,14,15]. Thus, understanding these changes over time and factors associated with PP decline is important. 

Longitudinal studies are scarce in middle and low-income countries in Latin America. Considering the sociodemographic effects on PP changes, additional studies in these countries are needed to elucidate and to direct public policies to help sustain the health of the older adults. Gait Speed (GS) is recognized as the best test to estimate overall health condition [1]. Although walking speed is the main component of the Timed Up and Go (TUG), it also assesses balance and strength, predicting the risk of falls [8,16]. Moreover, these tests are valid to measure the risk of negative outcomes such as physical disability, cognitive decline, falls, institutionalization, and mortality in older adults [1,2,3,4]. Therefore, this study evaluated changes in the GS and TUG time and associated factors among community-dwelling older adults in Brazil over five to six years of follow-up.

## 2. Materials and Methods

### 2.1. Study Population and Participant Recruitment

The study included participants of the Longitudinal Study of Older Adults Health—an ongoing longitudinal cohort study called “COMO VAI?”—Consórcio de Mestrado Orientado para Valorização da Atenção ao Idoso (Master’s Consortium Oriented for the Appreciation of Older Adults Care) initiated in 2014. Inclusion criteria were to be community-dwelling, age ≥ 60 years, and living in the urban area of the city of Pelotas, RS, Brazil (~330,000 inhabitants, 93% urban area) [17]. Those individuals unable to answer the questionnaire due to mental incapacity or who did not have a caregiver to assist them during the interview were not included in the study. For this study, the older adults who were unable to perform the PPTs in the 2014 and the 2019–2020 follow-up were also excluded. Of the 1451 older adults who met the inclusion criteria and agreed to participate in the study, the present study’s final sample included 476 older adults with available information on the GS and TUG in both periods (2014 and 2019–2020). The number of older adults recruited in 2014, as well as the follow-up rate, the identified deaths, and losses and refusals are presented in Figure 1.

The sample size was calculated based on the study of sarcopenia and frailty in older people from Pelotas. The sample size calculation estimated 1121 individuals, considering the prevalence of sarcopenia of 10%, 95% confidence interval, two percentage points as acceptable error and design effect of 1.10, plus 20% for losses and refusals [18]. For the study on frailty, the sample size calculation estimated 857 individuals, considering the prevalence of the outcome of 30%, 95% confidence interval, four percentage points as acceptable error and design effect 1.5, plus 20% for losses and refusals [19]. The sampling process were described elsewhere [18,19]. In brief, it took place in two stages: initially, 133 census tracts were randomly selected from the total of census sectors from Pelotas, based on data from the 2010 Brazilian Demographic Census [17]. In the second stage, 31 households were systematically selected per sector to enable the identification of at least 12 older adults in each one, based on a prior estimate of 0.43 older adult/household. It resulted in 1844 individuals being eligible to take part in the study.

Between January and August 2014, household interviews were carried out, in which a structured questionnaire was applied investigating general aspects related to the health of the older people, as well as sociodemographic variables. PPTs (GS and TUG) and anthropometric measurements were assessed by previously trained and standardized interviewers. In 2016–2017, a new phase of telephone/home interviews was carried out, in addition to monitoring mortality. From September 2019 to March 2020, a new and entirely home-based follow-up was carried out. In this follow-up, interviews were conducted and the PPTs measurements applied in 2014 were repeated. However, the third phase of the study needed to be interrupted due to the COVID-19 pandemic. From 900 individuals targeted to be interviewed in the 2019–2020 follow-up, 537 were actually followed. This study uses data from the first and third interviews of the participants of the “COMO VAI?” study.

The study was conducted according to the guidelines of the Declaration of Helsinki and approved by the Federal University of Pelotas Research Ethics Board (protocol code 472.357/2013 and 1.472.959/2016). Informed consent was obtained from all subjects involved in the study.

### 2.2. Assessment and Categorization of Physical Performance

PP was evaluated using GS and TUG. Both physical tests were performed twice, and the best performance was used for analysis. Walking aids were permitted if needed, but no caregiver assistance was allowed. GS was assessed by recording the time needed to walk a 4 m linear path without obstacles, using a stopwatch [2]. Before the test, individuals were instructed to walk at fastest speed, without running. The test was performed at the individual’s home on flat ground. Speed was calculated in m/s. TUG performance was assessed as the time in seconds the subjects took to rise from a chair, walk three meters without obstacles quickly but safely, turn around, walk back to the chair and sit down [16]. It was also measured using a stopwatch.

PPTs results were treated as continuous variables (m/s and s) according to time changes in PP tests from 2014 to 2019–2020. Clinically relevant changes were used to classify the performance into categories (declined, sustained, and improved) and they were defined as a variation ≥0.1 m/s in GS, considering previous studies [2,20], and ≥5% in TUG time. Higher TUG time and lower GS indicated PP decline.

### 2.3. Covariates at Baseline

The potential confounders accounted for in the analysis were age, sex, skin color (observed by the interviewer, considering that this is a marker of social inequality in Brazil), marital status, education level (based on years of education), socio-economic class (according to Associação Brasileira de Empresas de Pesquisa—ABEP [21]), current work status, diet quality (assessed using the Diet Quality Index for the Elderly [22]), leisure-time physical activity level (assessed by the International Physical Activity Questionnaire [23])—those who exercised at least 150 min/week were classified as active—, smoking history, alcohol consumption in the last month, multimorbidity (categorized into “up to four chronic diseases” or “five or more chronic diseases” [24]), presence of depressive symptoms (according to the Geriatric Depressive Scale—GDS-10 [25,26]), and polypharmacy (continuous use of five or more medications [27]). Subjects were classified as low weight, eutrophic, or overweight/obese based on age-specific cut-offs recommended by Lipschitz et al. [28] for Body Mass Index (BMI).

### 2.4. Statistical Analysis

Pearson’s chi-square test was used to assess possible differences between participants that completed or did not complete the follow-up. Differences in the mean GS and TUG time were analyzed using crude and hierarchically adjusted linear regression models. The variables that were associated with the exposure factor and the outcome were kept in the final adjustment model, considering a significance level of 20%. Associations between PP changes over time and sociodemographic, behavioral, and health variables were evaluated using mixed linear models. The adjustment model used was the same as for linear regression analysis. In all the models, the calculation of the second-order interaction between each exposure factor and the year in which the follow-up took place was included, to assess how much of the change in the PP was due to the analyzed exposure. All analyses were performed using Stata version 16.1 (Stata Corp., College Station, TX, USA). For all tests, a *p*-value < 0.05 was considered statistically significant.

## 3. Results

Figure 1 shows the study flow. Most of the participants were female (65.1%), with a mean age of 68 ± 6.7 years. Sociodemographic, behavioral, and health characteristics were similar in both assessments, except for a lower participation in the follow up of those aged 80 years or older at the first interview, and those who were widowers (Table 1).

Approximately 68% of the participants declined in GS and increased TUG time (declined PP). Almost 20% had no relevant change in GS, and 9% had no relevant changes in TUG time (sustained PP). Only 12% and 23% of the sample increased GS and decreased TUG time (improved PP), respectively (Appendix A).

The GS and TUG mean times in 2014 and 2019–2020 were 1.04 ± 0.36 m/s vs. 0.85 ± 0.32 m/s and 11.1 ± 8.3 s vs. 12.6 ± 8.0 s, respectively. Appendix A summarize the GS and TUG mean time values in both assessments, according to the independent variables.

Overall, there was a decline in PP. Being male (*p* = 0.023), living without a partner/separated (*p* = 0.035), higher education (*p* = 0.019), and alcohol consumption in the previous month (*p* = 0.045) were associated with GS decrease (Table 2 and Figure 2), while older age (*p* < 0.001), belonging to lower socioeconomic status (*p* < 0.004), physical inactivity (*p* = 0.017), and being overweight (*p* = 0.007) were associated with increased TUG time (Table 2 and Figure 3).

## 4. Discussion

Our findings showed that most participants presented a PP decline over the time of the study. Being male was the factor most strongly associated with GS decline and older age was the factor most strongly associated with TUG time increase. The PP decline among older people is a widely recognized phenomenon and can be attributed in part to aging-related declines in cardiovascular, musculoskeletal, and neuromuscular systems’ function, resulting in progressive loss of muscle mass and strength [2,3,6,8].

To the best of our knowledge, this is the first study to examine PP changes among community-dwellers in low- or middle-income countries according to clinically significant changes in PPTs, which makes it difficult to compare our results. However, a 4-year longitudinal study of 3018 community-dwelling older Chinese found an yearly decline in GS (−0.019 m/s/year in women and −0.025 m/s/year in men) over 4 years (−8.2% in men and −9.0% in women) [11]. Pinter et al. (2018) [29] evaluated the change in GS between the ages of 73 and 76 and found that 24% of the community-dwelling older people showed a significant clinical decline. The authors assumed the same measure (≥0.1 m/s) used in our study, but by year. Nevertheless, there is a growing number of studies evaluating PP trends or trajectories in different countries such as The Netherlands [7], Switzerland [12], Sweden [13], China [14], and Japan [15]. However, they require at least three measurements over time. Thus, both the definition of “low performance” and “meaningful decline” are variable across studies [30]. Some are based on sampling distributions (e.g., quartiles), others on external criteria (e.g., GS <0.8 m/s) and others on changes over time (e.g., ≥0.05 m/s) [29,31,32]. In our study we obtained similar results for both tests regarding the PP decline, in consonance with results from other studies using GS and TUG test [8,33,34].

On the other hand, an important proportion of participants exhibited a sustained or improved PP. Most of the participants (61%) were 60–70 years old in the baseline interview, which may explain the percentages of stability found in our study. The age between 80–85 is the period where the greatest changes in health are observed, affecting GS and TUG, as shown in previous studies [11,35,36]. This coincides with Lee et al. (2019), who observed stable trajectories in GS among community-dwelling older adults aged 65–84 years [36]. A PP increase may be observed due to changes in the participants’ lifestyle during this period. In this sense, numerous interventions regarding physical activity have shown progress in mobility, strength, and/or balance among healthy or community-dwelling older people [37,38]. Although both tests (GS and TUG) were used aiming to assess PP, these two tests evaluate different dimensions, since the TUG considers, in addition to mobility, strength and dynamic balance, and each of the dimensions is impacted in a different way with aging [7,36].

We assess PP through two widely recommended tests because they are simple, cheap, and fast measures, especially in community setting [8,33]. In our study, we observed that approximately a quarter of the participants (27%) did not have agreeing tests (data not shown), reinforcing the importance of using more than one PPTs in a complementary way to better target the necessary interventions among older people.

Our results showed that older age, being male, low socioeconomic status, higher education, living alone/separated, physical inactivity, being overweight, and alcohol consumption in the last month were associated with a PP decline. This set of factors, except higher education, is widely recognized in the literature and it is associated with worse indicators of overall health of older adults, including PP [9,10,30,31,39,40]. Considering that the PP decline is associated with numerous negative outcomes, which can be reduced by intervening on potentially modifiable risk factors, studies have shown consistent results related to the major influence of low physical activity and obesity on PP decrease [9,10,30,39,41,42]. In addition, obesity can coexist with lower muscle mass, favoring the emergence of a condition called sarcopenic obesity, negatively related to PP, explained by factors such as fatty infiltration in the muscles [43]. Regarding the level of education, older adults with higher education had a more pronounced decline in GS, probably due to the fact that they exhibited higher means both in 2014 and in 2019. In part, this can be explained by the cut-off point used to classify significant clinical change in GS, making it easier to identify variation in the faster ones and not in the slower ones. Previous study with U.S Chinese older adults found that higher education was one of the main factors associated with a faster rate of decline in all PP measures evaluated (including chair stand, tandem stand, and timed walk) [31].

Evidence points out that multicomponent training programs (MCT) are one of the most effective interventions to improve PP in older people [44]. An experimental study with 25 older adults showed that physical and cognitive status remained enhanced two years later compared to baseline, except for lower-limb strength. Among the evaluated physical functions, it was verified that strength and cardiovascular fitness were more sensitive to detraining, whilst agility proved to have larger training retentions [45]. A Spanish study observed positive effects of MCT in 106 older adults with or at greater risk of frailty, whereas 4-months of detraining caused a drop of variables related to functional capacity and frailty [46]. In this sense, the effects of detraining should be considered, although we emphasize the originality of our study, which aimed to understand how this occurs on a large scale, among community-dwelling older people in free-living, that is, exposed to common life habits (which include low practice of physical activity).

It is important to report that the most important exposures related to the PP decline are considered as non-modifiable factors, since advanced age and being male were the factors most strongly associated with TUG time increase and GS decline, respectively, in our cohort. As previously described, balance and strength are necessary to perform the TUG and it is known that balance worsens from the age of 70 and muscle strength reduces around 30% among individuals at this age [47]. Furthermore, advanced age represents an accumulation of intrinsic and extrinsic factors associated with aging that can contribute to the PP decline, as verified in a cross-sectional study among elders aged 80 years or older in Vietnam with GS assessed by 4-m walk test [48]. This finding is in line with other studies reporting PP declines in older adults [11,31,32,48], except for a recent study that found a meaningful increase in GS over 10 years of follow-up among over 1000 Swiss older adults [12]. Interestingly, this study evaluated whether new cohorts of older people are in better health than previous ones. The authors mention that the significant improvement observed in physical performance tests over the last decade may be a reflection of a new generation of individuals who enter into old age with better PP, probably due to a greater concern with health.

Even though men showed a more accentuated PP decline, they exhibited better performance in both tests and evaluations (2014 and 2019) than women: 1.16 vs. 1.03 m/s (baseline) and 0.90 vs. 0.83 m/s (follow-up) for GS, respectively, and 9.2 vs. 11.9 s (baseline) and 11.2 vs. 12.9 s (follow-up) for TUG time, corresponding to a greater percentage of variation in physical test measures between men (22.4% for GS and 29.3% for TUG) compared to women (19.4% for GS and 15.2% for TUG). Regarding the health-disease-disability process, men tend to have a worse general health condition, due to less access to health services and low adherence to the treatment of chronic conditions, which is reflected in lower life-expectancy [49]. On the contrary, studies have observed that GS decline was faster in women than in men [50,51]. This discrepancy may be due to differences in walking protocols (distance, usual pace, acceleration, and deceleration phase). Furthermore, Sialino et al. (2021) found that GS determinants were similar between genders, but the lower GS observed among women may be due to the fact that they have both greater sensitivity (higher BMI and lower level of physical activity) and greater exposure (low schooling, living alone, more chronic diseases) to GS determinants than men [52].

We highlight that this study has a longitudinal design and a robust statistical model of adjustment considering a large number of factors potentially associated with PP. In addition, the use of two widely recommended PPTs allows the comparison with populations from different countries. This study provides data on PP in community-dwelling older people, filling a gap related to the scarcity of research in Brazil and Latin America in this field.

Our study has several limitations. First, the assessment of PP was limited to two PPTs, leaving other important dimensions uncovered, such as specific measures of balance, endurance, and strength. Second, we considered a set of factors potentially associated with PP changes among older adults, as indicated in previous studies. However, other not investigated variables may contribute to the decline. Our group is planning another follow-up wave so that we can assess other factors, including the impact of the pandemic on PP. Third, the point used to determine a significant decline in GS can be troublesome, leading to classification errors. It may be easier to identify variation in the fastest but not in the slowest ones. For example, in individuals with normal GS at baseline (1.0 m/s), a decline of 0.1 m/s represents a variation of 10%. On the other hand, those with slow GS at baseline and who also presented 10% variation in the test were not classified with a decline in PP (from 0.80 m/s to 0.72 m/s, for example). For individuals with slow GS, 0.1 m/s represents a greater variation. However, it should be noted that our study used the value described in the literature for significant clinical change in PP [2,20]. Finally, the interruption of follow-up in 2020 due to the COVID-19 pandemic reduced participation of the older people; however, it did not affect the detection of significant associations in this sample [53].

Future studies evaluating other dimensions of PP will help to understand the changes related to the aging, in consonance with a recent study that observed little overlap between the trajectories of different PPTs, suggesting a combination of measures, especially in the assessment of younger older people [7]. Longitudinal studies evaluating PP changes in older people from the perspective assumed in this study are needed to identify an earlier PP decline, allowing preventive measures to be taken in a timely manner.

Our findings have practical implications. The high prevalence of PP decline reinforces the importance of including PPTs early in the health assessment routine of the older people aiming to better target interventions preventing mobility disorders, not only among those who already have low performance, but also among those that show decline. Primary care providers and other health professionals are important in early identification and timely intervention. Preventive clinical measures in all health care settings need to be engaged with public policies to allow community spaces that ensure healthy and active aging.

## 5. Conclusions

This study showed that PP declined for most participants. Being male was the factor most strongly associated with GS decline, and age was the factor most strongly associated with TUG time increase over a five- to six-year period. The high prevalence of PP decline over time signals the importance of including physical tests in yearly health assessments.

## Figures and Tables

**Figure 1 ijerph-20-05579-f001:**
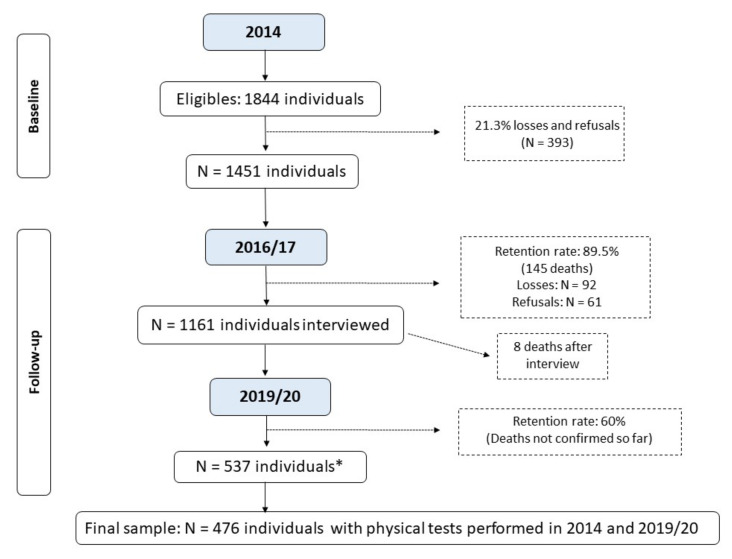
Flowchart of the Longitudinal Study of Older Adults Health: continuing the “COMO VAI?” study. * Number of living older adults located and interviewed before the study was interrupted due to the COVID-19 pandemic; the deaths that occurred until December/2022 were not yet verified by the epidemiological surveillance of Pelotas due to the duration of the pandemic.

**Figure 2 ijerph-20-05579-f002:**
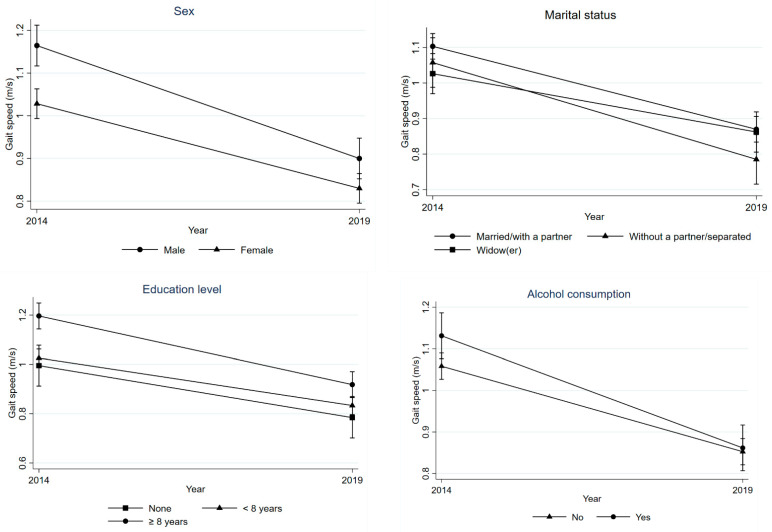
Difference in GS between 2014 and 2019 according to sex, marital status, education level, and alcohol consumption of individuals.

**Figure 3 ijerph-20-05579-f003:**
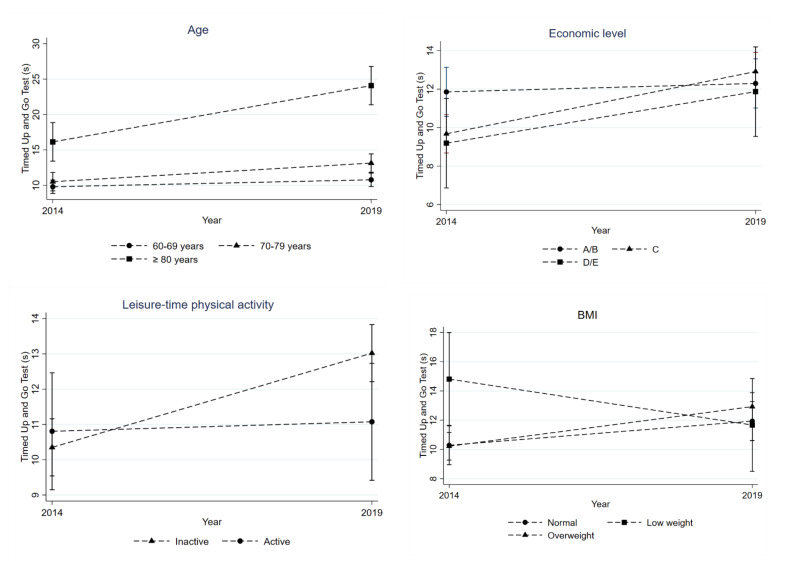
Evolution in terms of TUG time between 2014 and 2019 according to age, economic level, physical activity level, and body mass index (BMI).

**Table 1 ijerph-20-05579-t001:** Sample description according to sociodemographic, behavioral, and health-related characteristics in the 2014 and 2019 years of the COMO VAI? study.

	Complete Sample in 2014		Study Sample in 2019	
Variables	*n*= 1451	% (95%CI) ^a^	*n* = 476	% (95%CI) ^a^
Sex Female Male Age (completed years)	914 537	63.0 (60.5; 65.4) 37.0 (34.6; 39.5)	310 166	65.1 (60.7; 69.3) 34.9 (30.7; 39.3)
60–69 70–79 80+	756 460 230	52.3 (49.7; 54.9) 31.8 (29.5; 34.3) 15.9 (14.1; 17.9)	289 151 36	60.7 (56.2; 65.0) 31.7 (27.7; 36.1) 7.6 (5.5; 10.3)
Skin color				
White	1211	83.7 (81.7; 85.6)	389	81.7 (78.0; 85.0)
Other than white	236	16.3 (14.5; 18.3)	87	18.3 (15.0; 22.0)
Marital status				
Married/with a partner	763	52.7 (50.2; 55.3)	286	60.1 (55.6; 64.4)
Without a partner/separated	225	15.6 (13.8; 17.6)	72	15.1 (12.2; 18.6)
Widow(er)	459	31.7 (29.4; 34.2)	118	24.8 (21.1; 28.9)
Economic level ^b^				
A/B	483	35.2 (32.7; 37.8)	161	35.4 (31.1; 39.9)
C	720	52.5 (49.8; 55.1)	248	54.5 (49.9; 59.0)
D/E	169	12.3 (10.7; 14.2)	46	10.1 (7.7; 13.2)
Education level (completed years)				
None 1–7 ≥8	196 782 459	13.6 (12.0; 15.5) 54.4 (51.8; 57.0) 32.0 (29.6; 34.4)	54 270 151	11.4 (8.8; 14.6) 56.8 (52.3; 61.2) 31.8 (27.7; 36.1)
Current work situation				
No (unemployed)	1084	80.4 (78.2; 82.4)	344	77.0 (72.8; 80.6)
Yes (employed)	264	19.6 (17.6; 21.8)	103	23.0 (19.4; 27.2)
Diet quality ^c^				
Low	481	33.7 (31.3; 36.2)	143	30.2 (26.3; 34.6)
Average	534	37.5 (35.0; 40.0)	177	37.5 (33.2; 41.9)
High	411	28.8 (26.5; 31.2)	153	32.3 (28.3; 36.7)
Leisure-time physical activity (>150 min/week) ^d^				
No	1133	81.5 (79.3; 83.4)	378	80.4 (76.6; 83.8)
Yes	258	18.5 (16.6; 20.7)	92	19.6 (16.2; 23.4)
Smoking				
Not a smoker	781	54.0 (51.4; 56.6)	262	55.0 (50.5; 59.5)
Smoker	182	12.6 (11.0; 14.4)	58	12.2 (9.5; 15.5)
Former smoker	483	33.4 (31.0; 35.9)	156	32.8 (28.7; 37.1)
Alcohol consumption ^e^				
No	1138	78.8 (76.6; 80.8)	355	74.6 (70.5; 78.3)
Yes	307	21.2 (19.2; 23.4)	121	25.4 (21.7; 29.5)
Multimorbidity				
Up to 4 diseases	473	35.3 (32.8; 37.9)	175	37.6 (33.3; 42.1)
5 or more diseases	866	64.7 (62.1; 67.2)	291	62.4 (57.9; 66.7)
Depression ^f^				
No	1182	84.8 (82.8; 86.6)	408	86.4 (83.0; 89.3)
Yes	212	15.2 (13.4; 17.2)	64	13.6 (10.7; 17.0)
Polypharmacy ^g^				
No	513	35.6 (33.1; 38.1)	150	31.5 (27.5; 35.8)
Yes	929	64.4 (61.9; 66.9)	326	68.5 (64.2; 72.5)
BMI ^h^				
Low weight	126	9.2 (7.8; 10.9)	26	5.5 (3.8; 8.0)
Normal	471	34.5 (32.1; 37.1)	152	32.1 (28.1; 36.4)
Overweight	767	56.3 (53.6; 58.8)	295	62.4 (57.9; 66.6)

^a^ Pearson’s chi-square test. ^b^ According to Associação Brasileira de Empresas de Pesquisa (ABEP) [21]: category A/B indicating higher socioeconomic status; ^c^ assessed using the Diet Quality Index for the Elderly (Índice de Qualidade da dieta do idoso—IDQ-I) [22]; ^d^ assessed by the International Physical Activity Questionnaire (IPAQ) [23]; ^e^ alcohol consumption in the last month; ^f^ according to the Geriatric Depressive Scale (GDS-10) [25,26]; ^g^ continuous use of five or more medications [27]; ^h^ cut-offs recommended by Lipschitz et al. [28].

**Table 2 ijerph-20-05579-t002:** Gait Speed (GS) and Timed Up and Go (TUG) differences between the two assessments, according to sociodemographic, behavioral, and health-related variables. *n* = 476. Pelotas, Brazil.

Variables	GS *p*-Value ^a^ 2019–2014 Difference (95% CI)	TUG *p*-Value ^a^ 2019–2014 Difference (95% CI)
Sex Male Female Age (completed years)	*p* = 0.023 −0.26 (−0.31; −0.22) −0.20 (−0.23; −0.16) *p* = 0.88	*p* = 0.32 2.62 (1.19; 4.05) 1.71 (0.66; 2.76) *p* < 0.001
60–69 70–79 80+	−0.22 (−0.25; −0.18) −0.23 (−0.28; −0.18) −0.24 (−0.34; −0.14)	0.98 (−0.09; 2.05) 2.63 (1.16; 4.10) 7.95 (4.91; 10.98)
Skin color	*p* = 0.70	*p* = 0.20
White	−0.22 (−0.25; −0.19)	1.80 (0.85; 2.75)
Other than white	−0.23 (−0.30; −0.17)	3.26 (1.25; 5.26)
Marital status	*p* = 0.035	*p* = 0.35
Married/with a partner	−0.23 (−0.27; −0.20)	2.30 (1.19; 3.41)
Without a partner/separated	−0.27 (−0.34; −0.20)	2.86 (0.66; 5.06)
Widow(er)	−0.16 (−0.22; −0.11)	1.02 (−0.69; 2.74)
Economic level ^b^	*p* = 0.06	*p* = 0.004
A/B	−0.26 (−0.31; −0.22)	0.44 (−0.84; 1.72)
C	−0.21 (−0.24; −0.17)	3.23 (2.18; 4.29)
D/E	−0.16 (−0.25; −0.08)	2.68 (0.28; 5.08)
Education level (completed years)	*p* = 0.019	*p* = 0.14
None 1–7 ≥8	−0.21 (−0.29; −0.13) −0.19 (−0.23; −0.16) −0.28 (−0.33; −0.23)	3.87 (1.38; 6.35) 2.22 (1.10; 3.35) 1.01 (−0.50; 2.51)
Current work situation	*p* = 0.91	*p* = 0.45
No (unemployed)	−0.22 (−0.28; −0.17)	2.29 (1.36; 3.22)
Yes (employed)	−0.22 (−0.25; −0.19)	1.55 (−0.14; 3.24)
Diet quality ^c^	*p* = 0.82	*p* = 0.80
Low	−0.23 (−0.28; −0.19)	2.21 (0.78; 3.63)
Average	−0.22 (−0.26; −0.18)	2.44 (1.18; 3.71)
High	−0.21 (−0.26; −0.16)	1.81 (0.44; 3.19)
Leisure-time physical activity (>150 min/week) ^d^	*p* = 0.72	*p* = 0.017
No	−0.22 (−0.25; −0.19)	2.67 (1.80; 3.54)
Yes	−0.23 (−0.30; −0.17)	0.27 (−1.51; 2.04)
Smoking	*p* = 0.72	*p* = 0.93
Not a smoker	−0.21 (−0.25; −0.17)	1.92 (0.77; 3.08)
Smoker	−0.24 (−0.32; −0.17)	2.42 (−0.05; 4.88)
Former smoker	−0.23 (−0.28; −0.18)	2.17 (0.66; 3.68)
Alcohol consumption ^e^	*p* = 0.045	*p* = 0.11
No	−0.21 (−0.24; −0.17)	2.48 (1.48; 3.47)
Yes	−0.27 (−0.32; −0.22)	0.86 (−0.84; 2.56)
Multimorbidity	*p* = 0.69	*p* = 0.07
Up to 4 diseases	−0.22 (−0.26; −0.17)	1.21 (−0.08; 2.50)
5 or more diseases	−0.23 (−0.26; −0.19)	2.70 (1.71; 3.70)
Depression ^f^	*p* = 0.51	*p* = 0.89
No	−0.23 (−0.26; −0.20)	2.14 (1.30; 2.98)
Yes	−0.20 (−0.27; −0.13)	2.31 (0.23; 4.38)
Polypharmacy ^g^	*p* = 0.22	*p* = 0.34
No	−0.20 (−0.25; −0.15)	1.44 (−0.11; 2.98)
Yes	−0.23 (−0.27; −0.20)	2.34 (1.31; 3.38)
Body Mass Index ^h^	*p* = 0.33	*p* = 0.007
Low weight	−0.18 (−0.29; −0.06)	−3.14 (−6.71; 0.42)
Normal	−0.25 (−0.30; −0.20)	1.65 (0.16; 3.15)
Overweight	−0.21 (−0.25; −0.18)	2.71 (1.64; 3.78)

^a^ *p*-value obtained through adjusted mixed linear models including variables according to hierarchical levels: 1st Level: sex, age, skin color, marital status, economic level, education, work status; 2nd Level: diet quality, physical activity, smoking, alcohol consumption; 3rd Level: multimorbidity, depression, polypharmacy, and nutritional status. ^b^ According to Associação Brasileira de Empresas de Pesquisa (ABEP) [21]: category A/B indicating higher socioeconomic status; ^c^ assessed using the Diet Quality Index for the Elderly (Índice de Qualidade da dieta do idoso—IDQ-I) [22]; ^d^ assessed by the International Physical Activity Questionnaire (IPAQ) [23]; ^e^ alcohol consumption in the last month; ^f^ according to the Geriatric Depressive Scale (GDS-10) [25,26]; ^g^ continuous use of five or more medications [27]; ^h^ cut-offs recommended by Lipschitz et al. [28].

## Data Availability

The data used in this study can be obtained upon request from the corresponding author.

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
