# Peer review of "Changes in Physical Performance among Community-Dwelling Older Adults in Six Years"

_ijerph, 2023, doi:10.3390/ijerph20085579_

Round 1
Reviewer 1 Report
Dear Authors,
Above the well designed manuscript and research, this papper do not contribut to new information in this field.
You highlight the longitudinal design and a robust statistical model of adjustment considering a large number of factors potentially associated with PP.
This study provides data on PP in community-dwelling older people, filling a gap related to the scarcity of research in Brazil and Latin America on this
field.
Nevertheless, there are a lot of studies in Brazil with older people and with innovative strategies. Also analysing the detraining. So, this manuscript is well described but do not have new information.
Maybe if the authors add in the discussion some relative and important studies that focus on physical performance and training.
Author Response
Dear Reviewer,
On behalf of all authors, I thank you for reviewing and commenting on our article. A careful review of the article was carried out based on the comments pointed out by the reviewers and all suggestions for improvements were made as will be detailed below.
We would like to clarify that the literature review was conducted on the Pubmed and Lilacs platforms, covering international articles and with representativeness of research conducted in Latin America, aiming to identify longitudinal studies that evaluated physical performance (PP) as a primary outcome in the context of factors associated with changes in PP among community-dwelling older people. Despite identifying a growing interest in this topic in recent years, with Brazilian studies being located, we would like to emphasize that most of these were cross-sectional or experimental studies, with specific populations, such as institutionalized older people, patients with neurological diseases or other comorbidities, in outpatient/hospital or research contexts. Therefore, we reinforce that we did not find studies in Brazil or Latin America with longitudinal design that evaluated changes in PP in a representative sample of community-dwelling older adults with the perspective assumed in our study. In our understanding, evaluating PP based on significant clinical changes rather than cutoff points suggestive of low PP, in addition to providing data that allow estimating the magnitude of the problem and knowing potentially associated factors, fills an important gap in the literature and brings practical results, since it becomes possible to identify individuals who may benefit from preventive actions in a timely manner. For the suggestion to add in the discussion some relative and important studies that focus on PP and training, we included two experimental studies addressing the impact of detraining on older people (page 11), but we emphasize the originality of our study that aims to understand how this occurs on a large scale, with community-dwelling older people in free-living, that is, exposed to the common life habits of the older adults (which includes the low practice of physical activity). Thus, the focus of our study was not to evaluate detraining. Finally, we reviewed the references cited throughout the article and some references have been updated (3, 5, 17-19, 21, 23, 27, 46-56) and we rewrite the conclusion based on the results found in our study.
Please, find attached the revised version of the manuscript.
All authors have been approved the reviewed manuscript and agree with its submission to IJERPH.
Best regards,
Darlise Rodrigues dos Passos Gomes (Corresponding author)

Reviewer 2 Report
Manuscript is well written. Purpose is clear, and the introduction is well-written and supported with appropriate evidence.
Methods section could be more clearly explained. How was gait speed assessed? I am also confused about why interviews were performed in 2017 and what occurred in them. Please provide more detail in this section and thoroughly address the internal validity
Results: Use different types of lines in graphs for easier interpretation in black/white
Every question or concern is adequately answered in the discussion or Iimitation section.
Author Response
Dear Reviewer,
On behalf of all authors, I thank you for reviewing and commenting on our article.
Regarding your suggestions, in the methods section, we describe in more detail how the gait speed test was assessed (page 4), the follow-up carried out in 2017, sample size calculation and sampling corroborating the internal validity of the study (pages 2 and 3). Furthermore, we redraw the figures 2 and 3 using different types of lines in graphs for easier interpretation in black/white.
Please find attached the revised version of the manuscript
All authors have been approved the reviewed manuscript and agree with its submission to IJERPH.
Best regards,
Darlise Rodrigues dos Passos Gomes (Corresponding author)

Reviewer 3 Report
Section 2.1. Study Population and Participant Recruitment
The authors say: "The sample size and sampling process were described elsewhere [17,18]".
Comment: It is not stated how the sample was calculated.
Better describe inclusion and exclusion criteria.
Figure 1 should be improved. The flow chart is confusing. The figures do not match the final sample size.
The final sample size (476) should be clarified. How is this figure arrived at?
Section 2.3 Covariates at baseline
Delete the first paragraph in Portuguese.
Confounding factors include skin colour.
Comment: Not mentioned in the discussion. what value does it have? Delete or add reference.
Section 4 Discussion
The level of studies should be deepened with more information (if possible), as it is a striking result and should be compared with other studies.
Some references should be corrected and adapted to the Vancouver system correctly (e.g. 18, 21, 23, ...) should be cited with the abbreviation of the journal.
Author Response
Dear Reviewer,
On behalf of all authors, I thank you for reviewing and commenting on our article. A careful review of the article was carried out based on the comments pointed out by the reviewers and all suggestions for improvements were made as will be detailed below.
Regarding your suggestions, we reviewed the references cited throughout the article and some references have been updated (3, 5, 17-19, 21, 23, 27, 46-56), including the adaptation to the Vancouver system.
The sample size calculation as well as sampling and inclusion and exclusion criteria were detailed, corroborating the internal validity of the study (pages 2 and 3). Figure 1 was improved clarifying the final sample size.
In section 2.3 “covariates at baseline” the first paragraph in Portuguese has been deleted.
Regarding the variable “skin color”, included as a potential confounding factor, we clarify that it is an important indicator of inequality in Brazil, even though in our study we only classified "whites" and "non-whites" into two categories. There was no discussion in terms of results, given that this variable was not associated with physical performance tests.
In addition, the level of studies presented in the discussion was deepened with more information helping to compare with other studies.
Finally, we rewrite the conclusion based on the results found in our study.
Please, find attached the revised version of the manuscript.
All authors have been approved the reviewed manuscript and agree with its submission to IJERPH.
Best regards,
Darlise Rodrigues dos Passos Gomes (Corresponding author)

Round 2
Reviewer 1 Report
Dear Authors,
The manuscript is better. Good job.
Best regards.